# Polyherbal Combinations Used by Traditional Health Practitioners against Mental Illnesses in Bamako, Mali, West Africa

**DOI:** 10.3390/plants13030454

**Published:** 2024-02-04

**Authors:** Nastaran Moussavi, Pierre Pakuy Mounkoro, Seydou Mamadou Dembele, Nfla Ngolo Ballo, Adiaratou Togola, Drissa Diallo, Rokia Sanogo, Helle Wangensteen, Berit Smestad Paulsen

**Affiliations:** 1Section for Pharmaceutical Chemistry, Department of Pharmacy, University of Oslo, P.O. Box 1068, Blindern, 0316 Oslo, Norway; helle.wangensteen@farmasi.uio.no (H.W.); b.s.paulsen@farmasi.uio.no (B.S.P.); 2Faculty of Medicine and Odonto-Stomatology, University of Sciences, Techniques and Technologies of Bamako (USTTB), Bamako BP1805, Mali; 3Department of Traditional Medicine, National Institute of Public Health, Bamako PB1746, Mali; seydoumamadoudembele@yahoo.fr (S.M.D.); ngolo_ballo@yahoo.fr (N.N.B.); togola_adia@hotmail.com (A.T.); rosanogo@yahoo.fr (R.S.); 4Faculty of Pharmacy, University of Sciences, Techniques and Technologies of Bamako (USTTB), Bamako BP1805, Mali

**Keywords:** central nervous system, ethnobotany, ethnopharmacology, medicinal plants, *Securidaca longepedunculata*

## Abstract

This study explores the traditional knowledge of plants used by traditional health practitioners (THPs) in the treatment of symptoms or syndromes related to mental illnesses in the district of Bamako in Mali, along with the identification of affiliated traditional treating methods. An exploratory and cross-sectional ethnopharmacological survey was conducted in the district of Bamako. The Malian Federation of Associations of Therapists and Herbalists (FEMATH) assisted in the identification and inclusion of the THPs. Data sampling included semi-structured interviews, questionnaires, and in-depth interviews. Quantitative data were evaluated by analysing reports of the use of different medicinal plants and the number of participants. Fifteen THPs belonging to the district of Bamako participated. In total, 43 medicinal plants belonging to 22 plant families were used by the THPs. The most cited plant species was *Securidaca longepedunculata* (violet tree), followed by *Khaya senegalensis* (African mahogany) and *Boscia integrifolia* (rough-leaved shepherds tree). A great number of herbal combinations, preparation methods, and administration routes were used, often with honey as an adjuvant. To our knowledge, this is the first ethnobotanical survey on the use of medicinal plants in the treatment of all types of mental disorders in Bamako.

## 1. Introduction

The use of herbal combinations is part of traditional medicine in different cultures worldwide. The concept of combining two or more herbs in a formulation is based on a medicinal system that has existed since ancient times, often with the purpose of attaining better therapeutic efficacy and/or reduced profile of side effects, often favoured over single-herb treatment [1,2,3]. The pharmacological activity of herbs and plant extracts is often explained by the phenomenon of synergy and polyvalence. Synergy is defined within medicine as an effect where the combination of substances exhibits a greater effect than the sum of the effects of each individual substance [4,5]. While synergism only involves one specific type of interaction, polyvalence includes various forms of interactions also resulting in an overall enhancement of the pharmacological efficacy of the medicinal plants. On the other hand, the antagonistic effects of the herbs in a polyherbal drug formulation can also contribute to the favourable pharmacological effects. Therefore, inclusion of specific herbs, dosages, and the mixing ratio between the herbs cannot be arbitrarily made and are important for the pharmacological effects [3,6]. However, the mechanism of actions of many herbal medicines is still unknown. In Western medicine, the concept of therapy called ‘silver bullet’ or ‘one-drug-one-target’ is now increasingly regarded as inadequate in various clinical situations. Consequently, a combination of drugs is employed in the treatment of diseases such as cancer, HIV, and in palliative treatment [7,8]. Many plant-based drugs are used in Western evidence-based medicine, often named allopathic medicine. Examples of valuable allopathic plant drugs are quinine isolated from the *Cinchona* bark used to treat malaria [9], morphine isolated from the latex of *Papaver somniferum* L. (opium poppy) used to relieve severe pain [10], artemisinin obtained from leaves of *Artemisia annua* (Asteraceae) used as an antimalarial drug [11], podophyllotoxin isolated from roots, and rhizomes of *Podophyllum* species applied for the removal of condyloma (genital warts) [12].

Complementary and Alternative Medicine (CAM), known as integrated or holistic medicine, is defined as a heterogeneous set of health treatments and approaches applied together with, referring to ‘complementary’, or in place of, referring to ‘alternative’, a conventional model of medicine [13]. In general populations worldwide, the prevalence of all types of CAM used spans from 9.8% to 76% [14]. Clearly, the knowledge gap between the clinical relevance of CAM and Western scientific medicine is substantial [15]. Despite the extensive use of plants in the management of a variety of diseases in Mali, scientific studies on CAM and phytotherapy against mental disorders have been performed only to a small extent. One ethnobotanical survey has been published with a focus on traditional forms of treatment of schizophrenia spectrum disorders in Bandiagara, Mali [16].

A major cause of disease worldwide is attributable to mental disorders [17]. Nearly one billion people suffer from mental disorders. The World Health Organization (WHO) defines a mental disorder as a “clinically significant disturbance in an individual’s cognition, emotional regulation, or behaviour”. Mental disorders include schizophrenia, anxiety, depression, dissocial disorders, post-traumatic stress disorder, and neurodevelopmental disorders, such as autism spectrum disorder, and attention deficit hyperactivity disorder (commonly known as ADHD) [18]. In this paper, mental disorders, mental diseases, and mental illnesses may be referred to interchangeably. No data on the prevalence of mental disorders in Mali are available. However, it is suggested that the prevalence of mental disorders is comparable to other countries or even underestimated since Mali is a country with little or no access to treatment [19]. One study has identified that the prevalence of depressive and anxiety disorders in Mali is accountable for 3.6% and 2.6% of the estimated cases (crude prevalence rates, not age-standardized), respectively [20]. In most low- and middle-income countries, mental disorders remain both undertreated and underdiagnosed, even though the prevalence is high [19]. Estimates show that 76–85% of people with grievous mental disorders living in low- and middle-income countries receive no treatment [21]. The low mental-health workforce is concentrated in the capital city, Bamako. Here, only 0.03 psychiatrists are available per 100,000 inhabitants [22], whereas in comparison, 10.54 [23] and 20.91 psychiatrists are available in the US and France, respectively [24].

Traditional health practitioners (THPs) are often the first health care providers that inhabitants of developing countries approach when facing health challenges. In remote areas of Mali, approximately 80% of the population relies on traditional medicine for their primary health care. Inadequate modern healthcare systems and poor access to conventional medicines account primarily for the dependence on traditional medicine [19]. In this paper, THPs and healers may be referred to interchangeably.

This study aimed to explore and acquire knowledge about the traditional medical uses of formulations containing combined plants that are used by THPs in managing symptoms or syndromes related to mental illnesses by integrating all types of therapies in a complementary way, including traditional methods of treatment. It was performed in compliance with the objective of achieving optimal well-being through the development of evidence-based integrative medicine and psychiatry, applying all therapeutic providers and approaches available [25]. Interviews with healers relied on the common goal of improving human health through a better knowledge of plants used by the THPs.

## 2. Results and Discussion

### 2.1. Socio-Demographic Profile and Characteristics of Respondents

In total, 15 THPs belonging to the district of Bamako participated in the survey, all men. The THPs have no official degree in the study of traditional medicine. The Malian Federation of Associations of Therapists and Herbalists (FEMATH) ascertain their competence and assign their title as THP. This means that the FEMATH does not teach them to become THPs; their competence is being validated by the FEMATH. After assessing their abilities to treat a number of patients over a certain period of time with positive outcomes, they are considered approved as members of the association. All the THPs participating in this study were members of the FEMATH. The THPs’ knowledge about traditional medicine originated from their families, often transferred from father to son, while some were trained by other THPs and/or through revelation, meaning knowledge gained through their own experience, intuition, and/or God, see Table 1. Thirteen THPs were Bambara and two Dogon. Their average age was 51 years old. Nine THPs were illiterate and six were marabout. All considered themselves Muslims, whereas nine were the Quran’s teachers in addition to being THPs. Twelve had healing as their main profession and primary source of income, and two THPs specialized in mental health.

The THPs received on average 16 patients with mental illness per year. Three THPs included hospitalization of patients as part of the treatment; twelve did not due to lack of capacity at the hospitals. Eleven (73%) THPs communicated with academically trained physicians by exchanging information and referring patients as part of their patient treatment. In the academic comprehension of mental disorders, the THP’s local descriptions of causes of symptoms and syndromes can be covered with what we in Western medicine describe as mental illnesses. The reported symptoms and syndromes of the mental disorders fall within the internationally accepted ICD-10 Classification of Mental and Behavioural Disorders F00–F99 [26].

### 2.2. Plants Used and Similarity Level with the District of Bandiagara

In total, 43 medicinal and edible plant species belonging to 22 families were recorded during this survey (Table 2). Fabaceae was the most represented plant family, which included thirteen plant species, followed by Moraceae and Capparaceae, including three plant species each. The most used plant species was *Securidaca longepedunculata* (see Figure 1 and Figure 2). This plant was used both as the main plant and as an additive in polyherbal formulations. A minor fraction of the THPs (two) exploited *S. longepedunculata* as the only medicinal plant in the treatment of mental disorders, while nine THPs used this plant as the main species in combination with other medicinal plants (see Table 3). The THPs mainly used the roots and leaves of this plant, often as a fumigant or drink combined with other plants. Interestingly, *S. longepedunculata* was also the most used plant in an ethnobotanical study conducted in the district of Bandiagara, Mali. Here, the THPs used *S. longepedunculata* in the treatment of schizophrenia spectrum disorders [10]. The second and third most cited plants used as both main and additional plants were ascribed to *Khaya senegalensis* and *Boscia integrifolia*, respectively. Roots and bark of the trunk were the most frequent plant parts used by the THPs with regard to these two medicinal plants. When including and calculating all alternative plant parts reported per species regardless of the word “or” (i.e., “roots or leaves or bark of trunk” counted as three reports instead of one report), *Butyrospermum paradoxum* was positioned as the third most represented species, instead of *B. integrifolia*. Bulb of *Allium sativum* (eng: garlic) was the second most represented species when used as the main plant, cited by seven THPs, whereas the third most represented main plant was *K. senegalensis*, reported by six THPs. The roots, bark of trunk, and leaves of *K. senegalensis* were the most frequent plant parts used by the THPs.

As shown in Figure 1, all plant parts were used by the THPs for medicinal preparations: leaves, roots, fruits, bulbs, bark of trunk, bark of roots, resin, branches, and the whole plant, in addition to any roots or bark of the trunk of any comestible plant with one dead part and parasitic plants (often referred to as mistletoe by the THPs). Among the recorded plant parts, roots were the dominating part, followed by the bark of trunk and leaves. From the perspective of sustainable use of medicinal plants, root harvesting is more destructive to plants than the use of flowers, leaves, or buds [27]. As the ground anchoring part of the plant, roots are one of the major organs of a plant, absorbing water and nutrients, which is important for the growth and development of the living organism, and thus, roots are crucial for its survival [28]. Therefore, benign harvesting practices, such as collecting leaves instead of whole plants or roots, should be considered in order to protect the existence of the plant. Also, slow-growing plant species with limited abundances, such as *S. longepedunculata*, are potential subjects to resource exhaustion or extinction if not beneficial harvesting practices are formulated [16,27]. For this reason, several of the THPs in the district of Bamako mentioned that they rather use peripheral parts of the roots for medicinal purposes, instead of the main part of the root.

The percentage similarity (%) of the use of medicinal plants was calculated by comparing data from two districts in Mali. The total number of medicinal plants reported in Bamako (A = 43), the total number of medicinal plants used in Bandiagara (B = 41) [16], and the number of shared use of medicinal plants reported by the THPs in both Bamako and Bandiagara (C = 17) was taken into account. Based on presented data, the percentage similarity was identified to be 25%, which indicates low degree of overlapped use of medicinal plants in the treatment of mental diseases in these two areas.

In general, varying biodiversity, growing habitats, and access to plant species might explain the low similarity coefficient between districts. This might further lead to the creation of different ethnobotanical knowledge and/or different systems of medicine in these areas. Also, studies have shown that factors such as origin, sex, age, level of education, and wealth contribute to the formation of ethnobotanical knowledge [29]. Geographical isolation of ethnic groups or communities might also cause reduced knowledge transmission of medical uses of plants; hence, herein lies a possible explanation to the low degree of overlapped use of medicinal plants between the two districts. Comparative analysis of ethnobotanical knowledge across districts is not only important for identifying and preserving the knowledge of the THPs but is also useful in the discovery of new bioactive compounds and the development of new drug candidates [30]. The differences in flora between the districts seem to explain the low degree of overlap in the use of medicinal plants as Bandiagara and Bamako are situated in dissimilar regions geographically. For two–three decades, the most frequently used medicinal plant in both studies, *S. longepedunculata*, has reportedly been eradicated from the district of Bandiagara. The THPs in Bandiagara have reported finding their plant material in other areas such as Koutiala and Bamako instead. Changes in the natural access of *S. longepedunculata* might lead to the use of other medicinal plants by the THPs in the future, which exemplifies the importance of preserving the ethnobotanical knowledge of the THPs. Other species commonly used in both studies, *K. senegalensis*, *A. senegalensis*, and *B. senegalensis*, are also classified as endangered species [16]. This is the first time a comparative analysis between the two Malian districts, Bandiagara and Bamako, has been performed regarding the traditional use of medicinal plants on mental diseases.

### 2.3. Single and Polyherbal Formulations, Preparations, and Modes of Administration

In Table 3, each recorded plant is presented with information about the traditional use as the main medicinal plant associated with other plant species (or food), preparation methods, route of administration, and plant parts used by the THPs in the treatment of all types of mental disorders in the region of Bamako, Mali. The plant remedies were administered by the patient in the form of powder or liquid. The liquid form was obtained by maceration, infusion, or decoction. Modes of administration were drinking, fumigation, inhalation, steam, or body bath. In this study, the THPs defined fumigation as inhalation of smoke. The plant material was burned on a small fire so that the smoke could be inhaled by the patient. This process was employed only for the use of dry materials such as crushed or powdered plant materials. Inhalation of the medicinal plants included the use of powdered plant material which was reduced into a very fine powder and administered straight into the nose (without burning it). More precisely, inhalation of the medicinal plants was not defined as the administration of steam from boiled (fresh or dried) plant material where vapor of the hot decoction is inhaled by the patient [31].

**Table 3 plants-13-00454-t003:** Medicinal plants used in the treatment of mental disorders in the district of Bamako, Mali, presented with corresponding scientific and local plant names, preparation, and administration methods. In the column for “Main plant”, numbers are given to show that more than one healer used the same plant as the main plant.

Plant Species	Used Parts of Main Plant	Other Plant(s) in Combination *	Mode of Preparation	Mode of Administration
*Allium sativum*	1. Bulb	*Commiphora africana* (R)	Mixed, dried, and powdered	Fumigation
	2. Bulb	*Daniellia oliveri* (R) and *Maerua oblongifolia* (W)	Mixed, dried, and powdered	Fumigation
	3. Bulb	*Securidaca longepedunculata* (L and RB) and *Senna occidentalis* (L)	Mixed, dried, and powdered	Fumigation
	4. Bulb	*Commiphora africana* (R) and *Ricinus communis* (F)	Mixed, dried, and powdered	Fumigation
	5. Bulb	*Maerua oblongifolia* (L)	Powdered	Inhalation
	6. Bulb	*Securidaca longepedunculata* (PR) and honey	Decoction	Drinking
	7. Bulb	*Boscia senegalenis* (L), *Boscia integrifolia* (L and BT), *Calotropis procera* (L), and/or honey	Powdered	Fumigation
*Mangifera indica*	Bark of trunk	*Cochlospermum tinctorium* (RI), *Parkia biglobosa* (R or BT), *Khaya senegalensis* (R or BT), *Butyrospermum paradoxaum* (R or BT), *Annona senegalensis* (R or BT), *Boscia senegalensis* (R or BT), *Boscia integrifolia* (R or BT), *Securidaca longepedunculata* (R), *Sclerocarya birrea* (R or BT), *Daniellia oliveri* (R or BT), *Ficus iteophylla* (R), *Feretia apodanthera* (R), *Saba senegalensis* (R or SB), *Chamaecrista nigricans* (PR), any comestible plant with one part dead (R or BT), *Guiera senegalensis* (R or BT), *Terminalia macroptera* (R), and *Dichrostachys cinerea* (R)	Maceration	Drinking, body bath for one month
*Sclerocarya birrea*	Roots or bark of trunk	*Cochlospermum tinctorium* (RI), *Parkia biglobosa* (R and BT), *Khaya senegalensis* (R and BT), *Butyrospermum paradoxaum* (BT), *Annona senegalensis* (R and BT), *Boscia senegalensis* (R and BT), *Boscia integrifolia* (R and BT), *Securidaca longepedunculata* (R), *Daniellia oliveri* (R and BT), *Guiera senegalensis* (R), *Ficus iteophylla* (R), *Mangifera indica* (BR), *Feretia apodanthera* (R), *Saba senegalensis* (R and SB), *Chamaecrista nigricans* (PR), any comestible plant with one part dead (R and BT), *Terminalia macroptera* (R), and *Dichrostachys cinerea* (R)	Maceration	Drinking, body bath for one month
*Annona senegalensis*	Roots or bark of trunk	*Cochlospermum tinctorium* (RI), *Parkia biglobosa* (R or BT), *Khaya senegalensis* (R or BT), *Butyrospermum paradoxaum* (R or BT), *Guiera senegalensis* (R or BR), *Boscia senegalensis* (R or BT), *Boscia integrifolia* (R or BT), *Securidaca longepedunculata* (R), *Sclerocarya birrea* (R or BT), *Daniellia oliveri* (R or BT), *Ficus iteophylla* (R), *Mangifera indica* (BR), *Feretia apodanthera* (R), *Saba senegalensis* (R or SB), *Chamaecrista nigricans* (PR), any comestible plant with one part dead (R or BT), *Terminalia macroptera* (R), and *Dichrostachys cinerea* (R)	Maceration	Drinking, body bath for one month
*Uvaria chamae*	Leaves	*Borassus aethiopium* (L)	Decoction	Drinking, body bath, steam bath
*Calotropis procera*	1. Stem	*-*	Crushed, dried, and powdered	Fumigation
	2. Leaves	*Commiphora africana* (R and L), *Stylosanthes erecta* (W) and *Pterocarpus lucens* (W)	Crushed, dried and powdered	Fumigation
	3. Leaves	*Boscia senegalensis* (L), *Boscia integrifolia* (L and BT), *Calotropis procera* (L), *Allium sativum* (B) and/or honey	Powdered	Fumigation
*Saba senegalensis*	Roots or small branches	*Cochlospermum tinctorium* (RI), *Parkia biglobosa* (R or BT), *Khaya senegalensis* (R or BT), *Butyrospermum paradoxaum* (R or BT), *Annona senegalensis* (R or BT), *Boscia senegalensis* (R or BT), *Boscia integrifolia* (R or BT), *Securidaca longepedunculata* (R), *Sclerocarya birrea* (R or BT), *Daniellia oliveri* (R or BT), *Ficus iteophylla* (R), *Mangifera indica* (BR), *Feretia apodanthera* (R), *Chamaecrista nigricans* (PR), any comestible plant with one part dead (R or BT), *Terminalia macroptera* (R), *Guiera senegalensis* (R or BT), and *Dichrostachys cinerea* (R)	Maceration	Drinking, body bath for one month
*Cussonia arborea*	Hypertrophied branches extremity	*-*	Crushed and dried, decoction	Drinking
*Borassus aethiopum*	Leaves	*Uvaria chamae* (L)	Decoction	Body bath, drinking, steam bath
*Cochlospermum tinctorium*	Rhizome	*Parkia biglobosa* (R or BT), *Khaya senegalensis* (R or BT), *Butyrospermum paradoxaum* (R or BT), *Annona senegalensis* (R or BT), *Boscia senegalensis* (R or BT), *Boscia integrifolia* (R or BT), *Securidaca longepedunculata* (R), *Sclerocarya birrea* (R or BT), *Daniellia oliveri* (R or BT), *Ficus iteophylla* (R), *Mangifera indica* (BR), *Feretia apodanthera* (R), *Saba senegalensis* (R or SB), *Chamaecrista nigricans* (PR), any comestible plant with one part dead (R or BT), *Guiera senegalensis* (R or BT), *Terminalia macroptera* (R), and *Dichrostachys cinerea* (R)	Maceration	Drinking, body bath for one month
*Commiphora africana*	1. Resin	*Allium sativum* (CB)	Dried and powdered	Fumigation
	2. Resin	*Citrus orantifolia* (L and F)	Dried and powdered	Fumigation
	3. Resin	*Ricinus communis* (F) and *Allium sativum* (B)	Dried and powdered	Fumigation
	4. Resin and leaves	*Stylosanthes erecta* (W), *Pterocarpus lucens* (W), *Calotropis procera* (L), and/or honey	Crushed and dried, decoction	Fumigation
*Boscia integrifolia*	1. Leaves	*Boscia senegalensis* (L and BT), *Calotropis procera* (L), *Allium sativum* (B), and/or honey	Crushed, dried, and powdered	Fumigation
	2. Roots or bark of trunk	*Cochlospermum tinctorium* (RI), *Parkia biglobosa* (R or BT), *Khaya senegalensis* (R or BT), *Butyrospermum paradoxaum* (R or BT), *Annona senegalensis* (R or BT), *Boscia senegalensis* (R or BT), *Guiera senegalensis* (R or BT), *Securidaca longepedunculata* (R), *Sclerocarya birrea* (R or BT), *Daniellia oliveri* (R or BT), *Ficus iteophylla* (R), *Mangifera indica* (BR), *Feretia apodanthera* (R), *Saba senegalensis* (R or SB), *Chamaecrista nigricans* (PR), any comestible plant with one part dead (R or BT), *Terminalia macroptera* (R), and *Dichrostachys cinerea* (R)	Maceration	Drinking, body bath for one month
*Boscia senegalensis*	1. Leaves	*Boscia integrifolia* (L or BT), *Calotropis procera* (L), *Allium sativum* (B), and/or honey	Crushed, dried, and powdered	Fumigation
	2. Roots or bark of trunk	*Cochlospermum tinctorium* (RI), *Parkia biglobosa* (R or BT), *Khaya senegalensis* (R or BT), *Butyrospermum paradoxaum* (R or BT), *Saba senegalensis* (R or SB), *Boscia integrifolia* (R or BT), *Securidaca longepedunculata* (R), *Sclerocarya birrea* (R or BT), *Daniellia oliveri* (R or BT), *Ficus iteophylla* (R), *Mangifera indica* (BR), *Feretia apodanthera* (R), *Terminalia macroptera* (R), *Chamaecrista nigricans* (PR), any comestible plant with one part dead (R), *Guiera senegalensis* (R or BT), *Annona senegalensis* (R or BT), and *Dichrostachys cinerea* (R)	Maceration	Drinking, body bath for one month
*Maerua oblongifolia*	1. Whole plant	*Daniellia oliveri* (RE) and *Allium sativum* (B)	Dried and powdered	Inhalation
	2. Leaves and small branches	*-*	Decoction	Drinking, body bath
	3. Leaves	*Allium sativum* (B)	Powdered	Inhalation
*Guiera senegalensis*	1. Roots or bark of trunk	*Cochlospermum tinctorium* (RI), *Parkia biglobosa* (R or BT), *Khaya senegalensis* (R or BT), *Butyrospermum paradoxaum* (R or BT), *Annona senegalensis* (R or BT), *Boscia senegalensis* (R or BT), *Boscia integrifolia* (R or BT), *Securidaca longepedunculata* (R), *Sclerocarya birrea* (R or BT), *Daniellia oliveri* (R or BT), *Ficus iteophylla* (R), *Mangifera indica* (BR), *Feretia apodanthera* (R), *Saba senegalensis* (R or SB), *Chamaecrista nigricans* (PR), any comestible plant with one part dead (R or BT), *Terminalia macroptera* (R), and *Dichrostachys cinerea* (R)	Maceration	Drinking, body bath for one month
	2. Leaves	*-*	Crushed, dried, and powdered	Fumigation
*Terminalia macroptera*	Roots	*Cochlospermum tinctorium* (RI), *Dichrostachys cinerea* (R), *Khaya senegalensis* (R or BT), *Butyrospermum paradoxaum* (R or BT), *Saba senegalensis* (R or SB), *Annona senegalensis* (R), *Boscia integrifolia* (R or BT), *Sclerocarya birrea* (R or BT), *Ficus iteophylla* (R), *Mangifera indica* (BR), *Feretia apodanthera* (R), *Securidaca longipedonculata* (R or BT), *Chamaecrista nigricans* (PR), *Boscia senegalensis* (R), *Guiera senegalensis* (R or BT), *Daniellia oliveri* (R or BT), any comestible plant with one part dead (R), and *Parkia biglobosa* (R or BT)	Maceration	Drinking, body bath for one month
*Euphorbia balsamifera*	Bark of trunk	*-*	Crushed, dried, and powdered	Drinking
*Ricinus communis*	Fruits	*Commiphora africana* (RE) and *Allium sativum* (B)	Dried and powdered	Fumigation
*Senna occidentalis*	Leaves	*Securidaca longipedonculata* (L and RB) and *Allium sativum* (B)	Dried and powdered	Fumigation
*Acacia albida*	Bark of trunk	*-*	Crushed, dried, and powdered	Fumigation
*Afzelia africana*	Leaves, roots and bark of trunk	*-*	Dried and powdered	Drinking, body bath, inhalation
*Arachis hypogaea*	Seeds	*Khaya senegalensis* (BT)	Mixed, crushed, dried, and powdered	Fumigation
*Burkea africana*	Leaves	*Vitex madiensis* (L), *Bridelia ferruginea* (L), *Securidaca longipedonculata* (L), *Khaya senegalensis* (L), and *Tamaridus indica* (L)	Decoction	Drinking, body bath, steam bath
*Chamaecrista nigricans*	Powder of roots	*Cochlospermum tinctorium* (RI), *Parkia biglobosa* (R or BT), *Khaya senegalensis* (R or BT), *Butyrospermum paradoxaum* (R or BT), *Saba senegalensis* (R or SB), *Boscia integrifolia* (R or BT), *Securidaca longepedunculata* (R), *Sclerocarya birrea* (R or BT), *Daniellia oliveri* (R or BT), *Ficus iteophylla* (R), *Mangifera indica* (BT), *Feretia apodanthera* (R), *Terminalia macroptera* (R), *Boscia senegalensis* (R), any comestible plant with one part dead (R), *Guiera senegalensis* (R or BT), *Annona senegalensis* (R), and *Dichrostachys cinerea* (R)	Maceration	Drinking, body bath for one month
*Daniellia oliveri*	1. Roots or bark of trunk	*Cochlospermum tinctorium* (RI), *Dichrostachys cinerea* (R), *Khaya senegalensis* (R or BT), *Butyrospermum paradoxaum* (R or BT), *Saba senegalensis* (R or SB), *Annona senegalensis* (R), *Boscia integrifolia* (R or BT), *Securidaca longepedunculata* (R), *Sclerocarya birrea* (R or BT), *Ficus iteophylla* (R), *Mangifera indica* (BR), *Feretia apodanthera* (R), *Terminalia macroptera* (R), *Parkia biglobosa* (R, PR or BT), *Guiera Senegalenis* (R or BT), *Boscia senegalensis* (R) and *Chamaecrista nigricans* (PR)	Maceration	Drinking, body bath for one month
	2. Resin	*Maerua oblongifolia* (W), and *Allium sativum* (W)	Mixed, dried, and powdered	Inhalation
*Dichrostachys cinerea*	Roots	*Cochlospermum tinctorium* (RI), *Parkia biglobosa* (R), *Khaya senegalensis* (R or BT), *Butyrospermum paradoxaum* (R or BT), *Saba senegalensis* (R or SB), *Boscia integrifolia* (R or BT), *Securidaca longepedunculata* (R), *Sclerocarya birrea* (R or BT), *daniellia oliveri* (R or BT), *Ficus iteophylla* (R), *Mangifera indica* (BR), *Feretia apodanthera* (R), *Terminalia macroptera* (R), *Boscia senegalensis* (R), any comestible plant with one part dead (R), *Guiera senegalensis* (R or BT), *Annona senegalensis* (R), and *Chamaecrista nigricans* (PR)	Maceration	Drinking, body bath for one month
*Parkia biglobosa*	1. Roots or bark of trunk	*Cochlospermum tinctorium* (RI), *Dichrostachys cinerea (R)*, *Khaya senegalensis* (R or BT), *Butyrospermum paradoxaum* (R or BT), *Saba senegalensis* (R or SB), *Boscia integrifolia* (R or BT), *Securidaca longepedunculata* (R), *Sclerocarya birrea* (R or BT), *Daniellia oliveri* (R or BT), *Ficus iteophylla* (R), *Mangifera indica* (BR), *Feretia apodanthera* (R), *Terminalia macroptera* (R), *Boscia senegalensis* (R), any comestible plant with one part dead (R), *Guiera senegalensis* (R or BT), *Annona senegalensis* (R), and *Chamaecrista nigricans* (PR)	Maceration	Drinking, body bath for one month
	2. Seeds	*Securidaca longepedunculata* (RB)	Crushed, dried, and powdered	Fumigation
*Pterocarpus lucens*	Whole plant	*Commiphora africana* (RE and L), *Stylosanthes erecta* (W), *Calotropis procera* (L), and/or honey	Mixed, crushed, dried, and powdered	Fumigation
*Pterocarpus santalinoides*	Leaves	*Butyrospermum paradoxaum* (M)	Decoction	Drinking, body bath, steam bath
*Stylosanthes erecta*	Whole plant	*Commiphora africana* (RE and L), *Pterocarpus lucens* (W), *Calotropis procera* (L), and/or honey	Crushed, dried, and powdered	Fumigation
*Tamarindus indica*	Leaves	*Vitex madiensis* (L), *Bridelia ferruginea* (L) *Securidaca longipedonculata* (L), *Burkea africana* (L) and *Khaya senegalensis* (L)	Decoction	Drinking, body bath, steam bath
*Vitex madiensis*	Leaves	*Burkea africana* (L), *Securidaca longipedonculata* (L), *Khaya senegalensis* (L), *Bridelia ferruginae* (L), and *Tamarindus indica* (L)	Decoction	Drinking, body bath, steam bath
*Khaya senegalensis*	1. Roots or bark of trunk	*Cochlospermum tinctorium* (RI), *Dichrostachys cinerea* (R), *Daniellia oliveri* (R or BR), *Butyrospermum paradoxaum* (R or BT), *Saba senegalensis* (R or SB), *Annona senegalensis* (R), *Boscia integrifolia* (R or BT), *Securidaca longepedunculata* (R), *Sclerocarya birrea* (R or BT), *Ficus iteophylla* (R), *Mangifera indica* (BR), *Feretia apodanthera* (R), *Terminalia macroptera* (R), *Boscia senegalensis* (R), any comestible plant with one part dead (R), *Parkia biglibosa* (R or BT), *Guiera senegalensis* (R or BT), and *Chamaecrista nigricans* (R)	Maceration	Drinking, body bath for one month
	2. Leaves and bark of trunk	*-*	Decoction	Drinking, body bath
	3. Leaves	*Vitex madiensis* (L), *Bridelia ferruginea* (L) *Securidaca longipedonculata* (L), *Burkea africana* (L), and *Tamaridus indica* (L)	Decoction	Drinking, body bath, steam bath
	4. Leaves	*Securidaca senegalensis* (L)	Crushed, dried, and powdered	Fumigation
	5. Bark of trunk	*-*	Crushed, dried, and powdered	Fumigation
	6. Bark of trunk	*Arachis hypogeae* (SE)	Crushed, dried, and powdered	Fumigation
*Ficus platyphylla*	Bark of trunk	*-*	Decoction	Drinking, body bath
*Ficus sycomorus*	Parasitic plant	*-*	Decoction	Drinking, body bath
*Ficus thonningii*	1. Bark of trunk	*-*	Dried in the shade and powdered	Drinking, body bath, fumigation
	2. Roots	*Cochlospermum tinctorium* (RI), *Dichrostachys cinerea* (R), *Daniellia oliveri* (R or BR), *Butyrospermum paradoxaum* (R or BT), *Saba senegalensis* (R or SB), *Annona senegalensis* (R), *Boscia integrifolia* (R or BT), *Securidaca longepedunculata* (R), *Sclerocarya birrea* (R or BT), *Khaya senegalensis* (R or BT), *Mangifera indica* (BR), *Feretia apodanthera* (R), *Terminalia macroptera* (R), *Boscia senegalensis* (R), any comestible plant with one part dead (R), *Parkia biglibosa* (R or BT), *Guiera senegalensis* (R or BT), and *Chamaecrista nigricans* (PR)	Maceration	Drinking, body bath for one month
	3. Leaves	*-*	Decoction	Body bath, steam bath
*Bridelia ferruginea*	Leaves	*Vitex madiensis* (L), *Khaya senegalensis* (L), *Securidaca longepedunculata* (L), *Burkea africana* (L), and *Tamaridus indica* (L)	Decoction	Drinking, body bath, steam bath
*Securidaca longepedunculata*	1. Roots or bark of trunk	*Cochlospermum tinctorium* (RI), *Dichrostachys cinerea* (R), *Khaya senegalensis* (R or BT), *Butyrospermum paradoxaum* (R or BT), *Saba senegalensis* (R or SB), *Annona senegalensis* (R), *Boscia integrifolia* (R or BT), *Daniella oliveri* (R or BT), *Sclerocarya birrea* (R or BT), *Ficus iteophylla* (R or BT), *Mangifera indica* (BR), *Feretia apodanthera* (R), *Terminalia macroptera* (R), *Boscia senegalensis* (R), any comestible plant with one part dead (R), *Parkia biglibosa* (R or BT), *Guiera senegalensis* (R or BT), and *Chamaecrista nigricans* (PR)	Maceration	Drinking, body bath for one month
	2. Leaves or bark of roots	*-*	Dried in the shade and powdered	Drinking, body bath, steam bath
	3. Bark of roots and leaves	*Acacia albida* (BT)	Dried and powdered	Drinking, fumigation
	4. Leaves and bark of trunk	*-*	Decoction	Drinking, body bath
	5. Bark of roots and leaves	*Senna occidentalis* (L) and *Allium sativum*	Dried and powdered	Fumigation
	6. Leaves	*Burkea africana* (L), *Vitex madiensis* (L), *Khaya senegalensis* (L), *Bridelia ferruginae* (L), and *Tamirindus indica*	Decoction	Drinking, body bath, steam bath
	7. Powder of roots	*Allium sativum* (B) and honey	-	Drinking
	8. Leaves	*Khaya senegalensis* (L)	Crushed, dried, and powdered	Fumigation
	9. Bark of roots	*Parkia biglobosa* (SE)	Crushed, dried, and powdered	Fumigation
*Ziziphus mauritiana*	Fresh leaves	*-*	Crushed and maceration	Drinking, body bath
*Feretia apodanthera*	Roots	*Cochlospermum tinctorium* (RI), *Dichrostachys cinerea* (R), *Khaya senegalensis* (R or BT), *Butyrospermum paradoxaum* (R or BT), *Saba senegalensis* (R or SB), *Annona senegalensis* (R), *Boscia integrifolia* (R or BT), *Securidaca longepedunculata* (R), *Sclerocarya birrea* (R or BT), *Ficus iteophylla* (R), *Mangifera indica* (BR), *Terminalia macroptera* (R), *Boscia senegalensis* (R), any comestible plant with one part dead (R), *Parkia biglibosa* (R or BT), *Mangifera indica* (BT), *Daniellia oliveri* (R or BT), and *Chamaecrista nigricans* (PR)	Maceration	Drinking, body bath for one month
*Citrus × aurantiifolia*	Leaves and fruits	*Commiphora africana* (RE)	Dried and powdered	Fumigation
*Butyrospermum paradoxum*	1. Parasitic plant	*-*	Dried and powdered	Drinking, body bath
	2. Roots or bark of trunk	*Cochlospermum tinctorium* (RI), *Dichrostachys cinerea* (R), *Khaya senegalensis* (R or BT), *Feretia apodanthera* (R or BT), *Saba senegalensis* (R or SB), *Annona senegalensis* (R), *Boscia integrifolia* (R or BT), *Securidaca longepedunculata* (R), *Sclerocarya birrea* (R or BT), *Ficus iteophylla* (R), *Mangifera indica (BR)*, *Terminalia macroptera* (R), *Boscia senegalensis* (R), any comestible plant with one part dead (R), *Parkia biglibosa* (R or BT), *Guiera senegalensis* (R or BT), *Daniellia oliveri* (R or BT), and *Chamaecrista nigricans* (PR)	Maceration	Drinking, body bath
	3. Parasitic plant	*Pterocarpus santalinoides* (L)	Decoction	Drinking, body bath, steam bath
	4. Parasitic plant	*-*	Decoction	Drinking
	5. Parasitic plan	*-*	Decoction	Drinking
*Balanites aegyptiaca*	Bark of trunk	*-*	Crushed, dried in the sun, powdered, and decoction	Fumigation, drinking

* Plant part used in parenthesis; B = bulb, BR = bark, BT = bark of trunk, CB = crushed bulb, L = leaves, M = mistletoes (parasitic plant), PR = powder of roots, R = roots, RB = bark of roots, RE = resin, RI = rhizome, RT = roots of trunk, S = stem, SB = small branches, SE = seeds, W = whole plant, - = not applicable.

The majority of the THPs operated with combinations of many plants in order to treat mental disorders. The number of plants used by the THPs in various combinations ranged from 1 to 19 plants. Normally, the treatment of mental illnesses was either based on a small number of plant drugs (one to three medicinal plants) or a vast number of plant drugs combined (nineteen medicinal plants). In this study, a common combination of ten plants was often reported by the THPs, and these were named *S. longepedunculata*, *B. paradoxum*, *C. tinctorium*, *S. senegalensis*, *S. birrea*, *G. senegalensis*, *K. senegalensis*, *B. integrifolia*, *P. biglobosa*, and *B. senegalensis*. To our knowledge, limited information exists on the medical use of this plant combination by THPs in other countries. This complex mixture of medicinal plants is probably not randomly chosen, and further research is necessary in order to understand the contribution of the individual components, i.e., synergy, in such a complex blend of medicinal plants.

A comparison of the polyherbal formulations used by the THPs between the districts of Bandiagara and Bamako revealed some degree of commonality, see Table 2. Both districts shared the use of *S. longepedunculata* in combination with *S. birrea*, *P. biglobosa*, and *S. senegalensis*, while some of the THPs from the district of Bandiagara also included the medicinal plants *Lannea acida*, *Butyrospermum paradoxum*, and *Combretum micranthum*. Also, in the district of Bandiagara, *S. longepedunculata* was used by the THPs in another polyherbal formulation containing *A. sativum*, *Xylopia aethiopica*, *Zingiber officinale*, *Piper guineense*, and *Aframomum melegueta* in the treatment of schizophrenia spectrum disorders [16]. An ethnobotanical study of medicinal plants traditionally used in the rural Greater Mpigi region of Uganda revealed that the majority of the informants (53.3%) also use *S. longepedunculata* in mixtures with other medicinal plants, in monotherapy (20%) or both singly and in various combinations (26.7%), when prescribing herbal medicines. The specific plant species used in combination with *S. longepedunculata* were not described in this study [32]. In other African countries, *S. longepedunculata* has reportedly been used in combination with only one other ingredient. The Baganda tribe in the district of Kampala in southern Uganda uses powder of the roots from *S. longepedunculata* together with powder of the roots from *Steganotaenia araliacea* as a cold infusion to cure ascariasis. Also, roots of *S. longepedunculata* are mixed with roots of *Zanthoxylum zanthoxyloides*, where snuff or the smoke resulting from the burning roots is inhaled to treat malaria or fever [33]. A review study of plants used in divination in southern Africa has described the Chopi people in Mozambique using roots of *S. longepedunculata* in combination with *Sphedamnocarpus pruriens* for treating people believed to be possessed by evil spirits. In Zimbabwe, the same herbal formulation of powdered roots is blended in porridge and eaten for the treatment of epilepsy and convulsions and also washed with to “rouse the spirits” [34]. In south–central Zimbabwe, roots of *S. longepedunculata* are mixed with roots of *Annona stenophylla* and spread around the homestead as a snake repellent [35]. In addition, one ethnobotanical study from the rural Greater Mpigi region of Uganda has revealed the use of a few drops of lemon juice added to an aqueous decoction of leaves of *S. longepedunculata* in the treatment of sore throats. In this study, many of the informants used *S. longepedunculata* both in polyherbal formulations and in monotherapy when prescribing herbal medicines [32].

Several of the plants used by the THPs are known as poisonous plants [36,37]. *Annona senegalensis* is known for its toxic effects and used as a hunting poison. The root bark of this plant contains alkaloids and cyanogenic glycosides, which are believed to cause the poisonous effects. The latex from *Calotropis procera* contains toxic cardenolides and is highly irritating to the skin and mucous membranes. *Boscia* spp. are also recognized as hunting poisons, and two Boscia species, namely *B. integrifolia* and *B. senegalensis*, were used in this study. Further research is required to explore the toxic components in these plants. Both *Euphorbia balsamifera* and *Ricinus communis*, belonging to the Euphorbiaceae family, are notorious for their high toxicity. The latex from Euphorbia contains toxic diterpenes, while *R. communis* contains ricin, a lectin known as one of the most toxic compounds, in addition to the toxic alkaloid ricinine. The roots and trunk bark of *B. paradoxum* are poisonous, potentially due to the presence of saponins [36,37]. In this study, multi-herb formulations were often prepared by the THPs to create safer treatments. Sometimes, the THPs would combine more than fifteen different plants in a single oral dose to minimize the risk of toxic effects of certain plant(s) in the herbal medicine. A previous survey on toxic plants sold on the market in the district of Bamako revealed that the THPs added plants to counteract the toxicity of specific plants [38]. In that study, *S. longepedunculata*, *K. senegalensis*, and *D. oliveri* were described as toxic plants, which are plants also used by the THPs in our study. There are, however, no scientific studies that have investigated the toxic effects of these herbal mixtures. Toxicity studies that can give a measure of the safety profile for the preparations described are therefore highly warranted.

### 2.4. Combination of Medicinal Plants with Adjuvants

Adjuvants are defined by the WHO as supplementary substances that are added to herbal medicines “for the purpose of altering the pharmacological or therapeutic properties of the herbal materials, neutralizing or reducing toxicity, masking the taste, assisting formulation into suitable herbal dosage forms, maintaining stability or extending the storage time”. In herbal preparations, common adjuvants are, for example, water, honey, wine, milk, vinegar, and clarified butter [39]. In this study, honey was used in combination with the species *S. longepedunculata*, *C. procera*, or *A. sativum*. The THPs did not describe the use of non-plant-derived additions, such as honey, in combination with the medicinal plants any further except when combined with *C. procera*. The latex of the leaves of this shrub will cause irritation of the respiratory system when inhaled. In this study, honey was used in combination with *C. procera* to reduce this side effect. As an additive, honey might also make herbal preparations more palatable; the sweetness of honey might enhance or disguise the bitter flavour of medicinal plants. Historically, honey has been well-recognized in traditional medicine for centuries. The ancient Egyptians, Greeks, Assyrians, Romans, and Chinese used honey for the treatment of wounds and gut [40]. Today, honey is of proven value in treating bacterial infections [41]. Recently, one study has shown that honey (Tualang Honey) is a cognitive enhancer in schizophrenic patients when given as a supplement [42]. Many review studies describe promising in vitro effects of honey with regard to the cardiovascular system, as well as anti-cancer and gastrointestinal digestion activities, but recommend future research to aim for a deepened knowledge about the in vivo effects of honey [43,44,45]. Whether this non-plant-derived ingredient has an impact on the pharmacological activities of the abovementioned plants is, to our knowledge, relatively unexplored, but future studies should address this plant/non-plant combination on different cognitive domains and identify any scientific rationale with regard to the addition of honey.

### 2.5. Pharmacological Activities Based on Literature Surveys

The most represented species in this study, *S. longepedunculata*, is extensively and commonly used in Africa for various diseases and conditions (see Figure 2). In the literature, *S. longepedunculata* is used in the treatment of, for example, malaria, in Burkina Faso, Nigeria, and Kenya [46,47,48,49]; rheumatism [50], epilepsy, and as snake repellent in Zimbabwe [51]; neuropsychiatric disorders in Burkina Faso [52]; and abortion, fever, headache, tuberculosis, stomach pain, and possession of evil spirits in Nigeria [49]. In this study, the most used plant parts of *S. longepedunculata* are the roots and leaves, as shown in Figure 1. Most of the scientific research on *S. longepedunculata* has also focused on the roots. Although both in vitro and in vivo studies have been conducted on this plant, there is limited documented evidence on the activities of *S. longepedunculata* on the central nervous system (CNS). However, two in vivo studies on mice have revealed anticonvulsant and sedative properties of aqueous root extract of *S. longepedunculata* [53,54], as well as anxiolytic activities [53]. The presence of CNS activity of *S. longepedunculata* in these two studies might explain the use of this plant in the treatment of mental diseases in the district of Bamako, Mali. Furthermore, these studies indicate the relevance of further research on *S. longepedunculata* addressing the effects on mental diseases and conditions such as schizophrenia and psychosis.

**Figure 2 plants-13-00454-f002:**
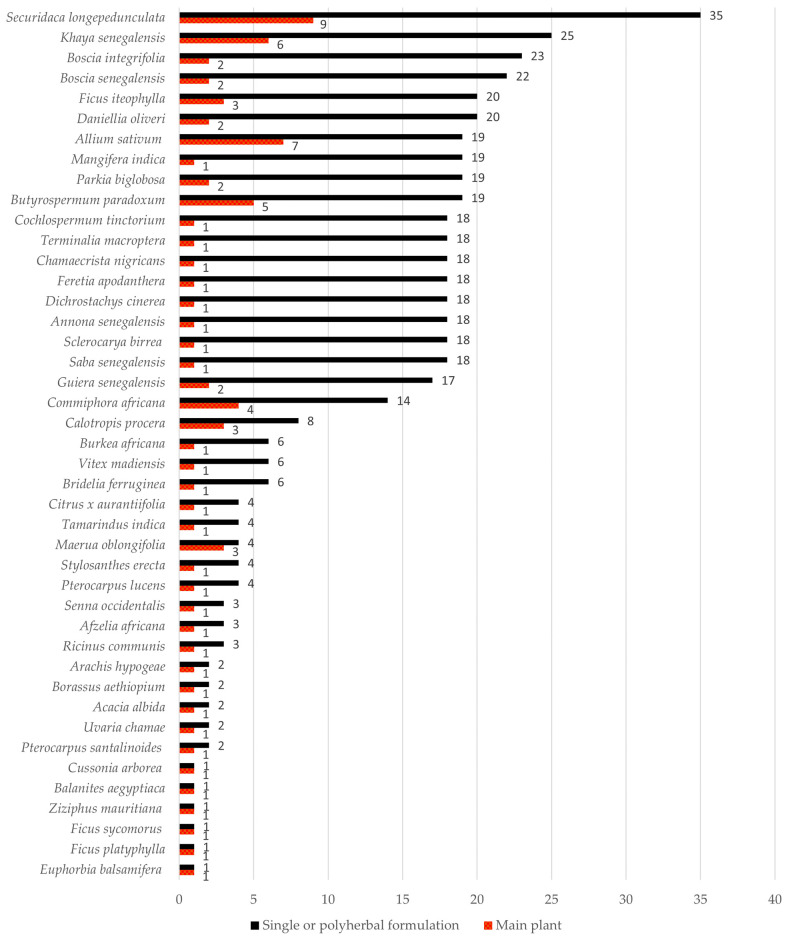
Total number of reports of plant species used for the treatment of mental illnesses in single or polyherbal formulations (black), and the total number of reports as the main plant for the treatment of mental illnesses (red) by the traditional health practitioners in Bamako, Mali.

*S. longepedunculata* is known to contain a series of xanthones [55], and the traditional medical use of this plant may be connected to these constituents. To our knowledge, no studies exist on the CNS effects of the xanthones reportedly found in *S. longepedunculata.* However, one in vitro study has shown that other xanthones, such as ɣ-mangosteen, garcinone, and gartanin, isolated from *Garcinia mangostana* are able to cross the blood–brain barrier (BBB) [56]. A parallel artificial membrane permeation assay (PAMPA) was used in order to evaluate the brain penetration of the xanthones. In addition, these three xanthones have been described as biologically active in anti-Alzheimer assays, possessing inhibitory effects against Aβ42 in *E. coli* cells, towards Aβ42 self-induced aggregation in a tube, and against BACE1. One Chinese patent of 6-hydroxy-1,2,3,7-tetramethoxyxanthone has been shown to improve depression through the increased number of hippocampal neural stem cells [57]. These research findings, together with the knowledge of xanthones being slightly lipophilic [58], could be promising for the xanthones isolated from *S. longepedunculata* in the therapy of mental illnesses. However, one in silico study of 2,6,7,8-tetramethoxyxanthone and 3-hydroxy-2,6,7,8-tetramethoxyxanthone predicted their BBB penetration rate as low [59]. These various results indicate the need for further studies on the CNS effects of extracts, xanthones, and other constituents from *S. longepedunculata*.

## 3. Materials and Methods

### 3.1. Study Area

Bamako is the capital and largest city of the Republic of Mali situated by the Niger river. The name Bamako originates from the Bambara word meaning “crocodile river”. With an estimated population of 2.81 million in 2022, the district is home to a variety of ethnic groups from both Mali and neighbouring countries. The relatively flat area of Bamako is divided into six communes with the following geographic coordinates: 12°38′21″ N 8°0′10″ W. In the nation’s administrative centre, 70% of the industrial activity in the country is concentrated in the district of Bamako, together with the presence of Bozo fishermen, craftsmen, and commerce. Women represent 49.8% of the population in this area. In addition to two hospitals and one paediatric hospital in Bamako, a new hospital called “Hôpital du Mali” was built in 2012. French is the official language of Mali. However, the Mandé language, Bambara/Bamanankan, is the most spoken language in Bamako and the main mother tongue in Mali (46%). Mali has a hot desert climate in the northern part, where the sandy and rocky land of Dogon (Bandiagara) is found, while the southern areas, where the district of Bamako is situated, are wet between July and September and dry the rest of the year.

### 3.2. Ethnopharmacological Data Collection

In the period of August to October 2017, an exploratory and cross-sectional ethnopharmacological survey on mental disorders, investigated as a collective term, was conducted in the district of Bamako. Ethnopharmacological data were collected using qualitative research methods. Methods employed in the data sampling included semi-structured interviews, questionnaires (face-to-face), and in-depth interviews [60]. The reason for exclusively focusing on medicinal plants in the treatment of mental illnesses and for choosing the district of Bamako was that one of the authors, unfortunately now deceased, was a medical specialist in mental illnesses and previously no investigation had been performed in Bamako related to these illnesses. Information about local plant names, plant parts used, modes of preparation, routes of administration, therapeutic methods and tools, number of daily doses, approximate amounts of the plant and duration of the treatment, origin of their knowledge, and traditional interpretations of mental illnesses were collected using a questionnaire with inquiries of the ethnobotanical practices of medicinal plants, (see Appendix A). The questionnaire for interviewing the healers included questions that made it possible to trace each healer if necessary. FEMATH assisted in the identification and inclusion of 15 THPs from the district of Bamako. In this study, the two languages, Bambara/Bamanankan and Dogon, were used for interviews and questionnaires. Two Dogon dialects were spoken; Tommo-soo and Tomokan. All THPs were members of an association of healers, and two of them were in the position of presidents of their associations. Prior to the interviews, individual appointments were made with each THP to make sure that the THP agreed to take part in the survey. All THPs were interviewed separately, and the interviews took place where they lived. Authors of this study were knowledgeable in one or both languages, Bambara and Dogon, as well as the two aforementioned dialects. The inclusion of extra interpreters providing language help was therefore unnecessary. In order to help the THPs understand the definition of mental disorders, different characteristics of the various disorders were presented and exemplified. All the characterizations of symptoms and syndromes described by the THPs were registered as mental disorders, not as specific and unique mental illnesses. Any of the practitioner’s descriptions of different types of local symptoms and syndromes that showed some similarity to the academic concept of mental disorders were registered. The semi-structured questionnaire was designed with regard to the current literature on mental disorders, which made it possible to enumerate the local syndromes and symptoms in kinship with the academic concept of mental disorders.

### 3.3. Plant Collection and Identification

The plant material was harvested in the village and region of Siby located about 50 km northwest of Bamako. Since the relevant medicinal plants grow in this area, herbalists (sellers of medicinal plants) and THPs usually travel this distance to collect them. In this study, the plants were collected by the THPs, and the macroscopic identification of each medicinal plant was conducted by two qualified botanists, both part of the research team, representing the Department of Traditional Medicine (DMT) in Bamako, Mali. The described plant material was collected for the documentation and, thereby, preparation of herbarium voucher specimens assuring the identification of the plants. The voucher specimens were deposited in the DMT herbarium in Bamako. Herbarium numbers of each plant, together with their vernacular names, are shown in Table 2. Scientific plant names and corresponding family names were checked and updated by means of The World Flora Online (http://www.worldfloraonline.org/ (accessed on 11 April 2023)).

### 3.4. Data Analysis for Similarity Level

An ethnopharmacological study on mental diseases conducted in the district of Bandiagara [16] was included in this comparative analysis to find the overlapping portion of medicinal plants used in our study and Bandiagara (also called Dogonland). It must be emphasized that the study performed in Dogonland focused on the use of medicinal plants specifically in the treatment of the mental disease schizophrenia, while our study included all mental illnesses. However, the comparison between these two regions in Mali is still valuable because it might detect medicinal plants that stand out as potential drug candidates in the treatment of mental illnesses. The percentage similarity of the diverse use of medicinal plants registered in both studies was calculated as follows:$$\mathrm{Percentage}\;\mathrm{similarity}\;(\%)=\;\frac{C}{A+B-C}\times 100$$
where A is the total number of species used in Bamako, B represents the total number of species used in Dogon [16], and C is the number of species used in both districts.

### 3.5. Ethical Considerations

Written informed consent was obtained from all the THPs prior to the interview, and all data were analysed anonymously. These documents were deposited at the DMT. Gifts were bestowed upon each THP as compensation for their time and effort in this study. The study was conducted in accordance with international, national, and institutional rules concerning the biodiversity rights.

## 4. Conclusions

This is the first study of its kind focusing on the indigenous use of medicinal plants for the treatment of all types of mental illnesses in Bamako. According to the aim of this study, a comprehensive overview of plant-based therapies for the treatment of mental disorders in the district of Bamako is offered. The most commonly used plant species was *Securidaca longepedunculata*, while the second and third most cited plants were *Khaya senegalensis* and *Boscia integrifolia*, respectively. The coherence in the use of medicinal plants for the treatment of mental disorders mainly involved the use of roots and leaves of *S. longepedunculata*, which was frequently used both as the main plant and in association with other plants, often as a fumigant. Honey was often combined with *S. longepedunculata* in the polyherbal formulations. Various modes of preparation of the plants were applied in the treatment of mental disorders, such as the preparation of maceration and decoction. However, one specific plant-based treatment with *S. longepedunculata* was not disclosed as each of the THPs used a variety of plant combinations and different techniques. The effects of honey as an adjuvant in herbal medicines should be investigated in future studies. This study follows up the recommendation from the Mounkoro group [16] to compare the traditional use of medicinal plants in the different districts, Bandiagara and Bamako, in Mali, in which our analysis showed a low degree of similarity in the use of medicinal plants in the treatment of symptoms and syndromes related to mental illnesses by the THPs. Future research should also study the THPs’ use of medicinal plants in relation to specific mental illnesses, instead of mental illnesses in its wholeness.

## Figures and Tables

**Figure 1 plants-13-00454-f001:**
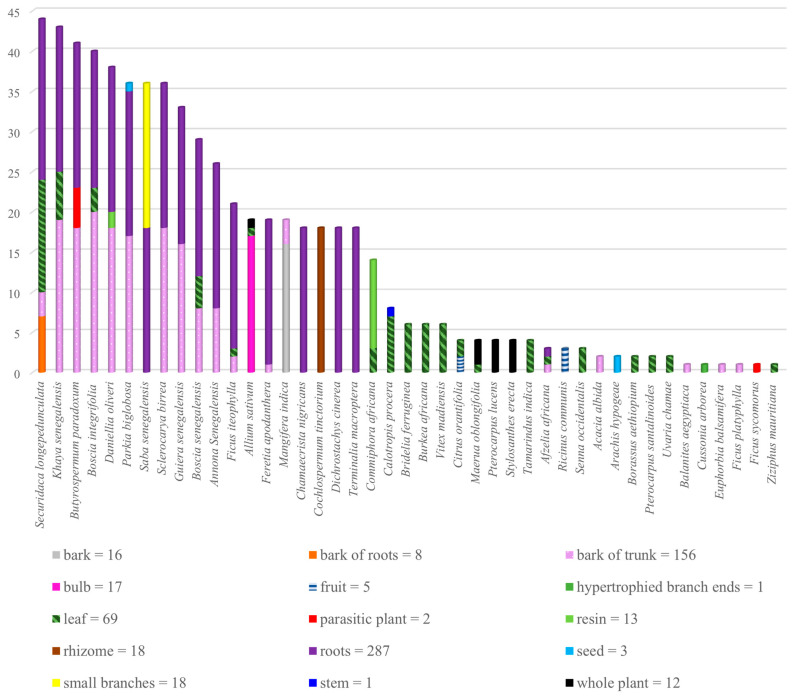
The number of reported plant parts used per species in the treatment of mental disorders by the traditional health practitioners (THPs) in Bamako, Mali. All plant parts are included without any differentiation between the words “and” or “or” reported by the THPs, cf. Table 3. The total number of uses for each plant part is shown below the diagram.

**Table 1 plants-13-00454-t001:** Origin of the traditional practitioners’ knowledge in the district of Bamako, Mali.

Origin of Knowledge	Number of Times Reported	Percentage
Family	3	20
THP(s) (not family)	3	20
Family and THP(s)	6	40
Family, THP(s) and revelation	3	20
Total	15	100

**Table 2 plants-13-00454-t002:** Scientific plant names and corresponding family names, herbarium numbers, and vernacular names of medicinal plants used against mental illnesses by the traditional health practitioners (THPs) in Bamako, Mali. In addition, the same medicinal plants used by THPs in both the districts of Bandiagara and Bamako, Mali, are shown. The order of family names is given alphabetically relative to the plant species. Hyphen is marked in the table when no information given.

Plant Species (Common Synonyms)	Family Name	Plant Names in Local Languages			Herbarium Number	Used in Bandiagara, Mali for the Treatment of Schizophrenia [16]
		Bambara/ Bamanankan	Dogon			
			Tomokan	Tommo-Soo		
*Allium sativum* L.	Amaryllidaceae	Layi	-	-	2814/DMT	x
*Mangifera indica* L.	Anacardiaceae	Mangoro	-	-	1467/DMT	-
*Sclerocarya birrea* (A.Rich) Hocsht	Anacardiaceae	Gounan	-	-	0071/DMT	x
*Uvaria chamae* P.Beauv	Annonaceae	Toufing	-	-	30697/DMT	-
*Annona senegalensis* Pers.	Annonaceae	Mande sunssun	-	-	0281/DMT	x
*Calotropis procera* (Aiton) W.T.Aiton	Apocynaceae	Fogofogo	-	-	2901/DMT	-
*Saba senegalensis* (A.DC) Pichon	Apocynaceae	Zaban	-	-	1252/DMT	x
*Cussonia arborea* Hochst. Ex A.Rich	Araliaceae	Bolokourounin	-	-	2642/DMT	-
*Borassus aethiopum* Mart.	Arecaceae	Sébé	-	-	No registration	-
*Cochlospermum tinctorium* Perrier Ex A.Rich	Bixaceae	N’tiribala	-	-	2298/DMT	-
*Commiphora africana* (A.Rich) Engl.	Burseraceae	Barakanté	-	-	249/CRMT	x
*Boscia integrifolia* J.St.-Hil. (syn. *Boscia angustifolia* A.Rich.)	Capparaceae	Tchiè béré	Koulale-toumbo	-	0980/DMT	x
*Boscia senegalensis* Lam.	Capparaceae	Musso béré	Tougoutale	-	0735/DMT	x
*Maerua oblongifolia* (Forssk.) A.Rich.	Capparaceae	-	-	-	No registration	-
*Guiera senegalensis* J.F. Gmel (syn. *Guiera glandulosa* Sm.)	Combretaceae	Goundiè	-	-	0537/DMT	x
*Terminalia macroptera* Guill. & Perr	Combretaceae	Wôlô	-	-	3752 /DMT	-
*Euphorbia balsamifera* Aiton	Euphorbiaceae	-	Sindi	-	353/DMT	x
*Ricinus communis* L.	Euphorbiaceae	Tomotigui	-	-	0235/DMT	-
*Senna occidentalis* (L.) Link	Fabaceae	N′palanpalan	-	-	1525/DMT	-
*Acacia albida* Delile (syn. *Faidherbia albida*)	Fabaceae	Balazan	-	-	0495/DMT	x
*Afzelia africana* Sm. ex Pers	Fabaceae	Lingue	-	-	3038/DMT	-
*Arachis hypogaea* L.	Fabaceae	Tiga	-	-	2885/DMT	-
*Burkea africana* Hook.	Fabaceae	Siri	-	-	3756/DMT	-
*Chamaecrista nigricans* (Vahl) Greene	Fabaceae	Dialanin Tchèma	-	-	1524/DMT	-
*Daniellia oliveri* (Rolfe) Hutch. & Dalziel	Fabaceae	Sana	-	-	0909/DMT	x
*Dichrostachys cinerea* (L.) Wight & Arn.	Fabaceae	Giriki or Goron	-	-	2839/DMT	-
*Parkia biglobosa* (Jacq.) R.Br. ex G.Don (syn. *Parkia biglobosa* (Jacq) G.Don)	Fabaceae	Nêrê	-	-	1801/DMT	x
*Pterocarpus lucens* Lepr. ex Guill. & Perr.	Fabaceae	-	-	Tabagala	1090/DMT	-
*Pterocarpus santalinoides* L’Hér. ex DC.	Fabaceae	Nyeekoun	-	-	0092/DMT	-
*Stylosanthes erecta* P. Beauv	Fabaceae	-	-	Yagayaga	1092/DMT	-
*Tamarindus indica* L.	Fabaceae	N’tomi	-	-	2049/DMT	-
*Vitex madiensis* Oliv	Lamiaceae	Koroninfin	-	-	3019/DMT	-
*Khaya senegalensis* (Desv.) A.Juss	Meliaceae	Jala	-	-	2257/DMT	x
*Ficus platyphylla* Delile	Moraceae	N’gababilen	-	-	2332/DMT	x
*Ficus sycomorus* L.	Moraceae	Toroba	-	-	0139/DMT	-
*Ficus thonningii* Blume (syn. *Ficus iteophylla* Miq)	Moraceae	Jatiguifaga-yiri, Nsêrênindjê	-	-	1808/DMT	x
*Bridelia ferruginea* Benth	Phyllanthaceae	Sagan	-	-	0148/DMT	-
*Securidaca longepedunculata* Fresen.	Polygalaceae	Diro or Djoro	-	-	2220/DMT	x
*Ziziphus mauritiana* Lam	Rhamnaceae	N’tomono	-	-	2223/DMT	-
*Feretia apodanthera* Delite	Rubiaceae	Djoulasounkalani or mourlan	-	-	1944/DMT	-
*Citrus × aurantiifolia* (Christm.) Swingle	Rutaceae	Lenbourou	-	-	2250/DMT	-
*Butyrospermum paradoxum* (C.F.Gaertn.) Hepper (syn. *Vitellaria paradoxa* C.F.Gaertn.)	Sapotaceae	Sii	Mindji	-	0138/DMT	x
*Balanites aegyptiaca* (L.) Delile	Zygophyllaceae	Zèguenè	-	-	3074 /DMT	-

## Data Availability

Data are contained within the article and Appendix A.

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
