# Peer review of "Polyherbal Combinations Used by Traditional Health Practitioners against Mental Illnesses in Bamako, Mali, West Africa"

_plants, 2024, doi:10.3390/plants13030454_

Round 1
Reviewer 1 Report (Previous Reviewer 1)
Comments and Suggestions for Authors
1- Please, type keywords according to the alphabetic order.
2- It is suggested to use between 3-5 words for keywords, and it is better to use keywords different from the words which have been used in Title.
3- Referencing system in the manuscript has not done properly for example (1,2,3) should change into (1-3).
4- Correct and change plant(s), effect(s), and herb(s) to plants, effects and herbs.
5- It is suggested that authors use English names of medicinal plants and type scientific names inside parenthesis.
6- It is suggested that authors make the introduction shorter.
7- Abbreviation should be done once, for example when you use THPs in Abstract, you do not need to repeat is in introduction (line 96). After the first full title and words, authors should use the abbreviation form.
8-It seems Table 2 is not on the basis of the format of journal. Please, check all tables and figures.
9- In some parts of the manuscript, paragraphing is not clear at all, for example in section 2.3 and other parts of the manuscript. Each paragraph should be started with new point and content. Please, revise it carefully.
10- Table 3 does not prepared on the basis of journal s format.
11- Please, check the format of reference 57.
12- All names of authors should be write for reference 56.
13- n in reference 56 should big letter.
14- Please, check the format of all references.
15- Some references do not have DOI. Please, check them again that if they have DOI add it in the Reference section.
Other sections of the article is OK. The article can be accepted after considering these suggestions.
Author Response
Reviewer 1:
1- Please, type keywords according to the alphabetic order.
The keywords are now listed in alphabetic order.
2- It is suggested to use between 3-5 words for keywords, and it is better to use keywords different from the words which have been used in Title.
We have reduced the number of keywords and deleted those that are similar with the title.
3- Referencing system in the manuscript has not done properly for example (1,2,3) should change into (1-3).
We have changed the referencing system according to the reference style.
4- Correct and change plant(s), effect(s), and herb(s) to plants, effects and herbs.
Done
5- It is suggested that authors use English names of medicinal plants and type scientific names inside parenthesis.
In this article the latin names are used with intention to avoid any misidentifications.
6- It is suggested that authors make the introduction shorter.
Parts of the introduction has been deleted to shorten the introduction.
7- Abbreviation should be done once, for example when you use THPs in Abstract, you do not need to repeat is in introduction (line 96). After the first full title and words, authors should use the abbreviation form.
We believe that the abstract is independent of the rest of the manuscript. If THPs was abbreviated in the abstract, it needs to be described with words in the main text, then followed by the abbreviation. We have carefully inspected the whole document and we think this is correct now.
We find the same format for abbreviations in several recent papers in Plants, e.g. Alenazi MM, et al. Plants 2024, 13(3), 398; Sui D, et al. Plants 2024, 13(3), 397; Beca-Carretero P, et al. Plants 2024, 13(3), 396.
8-It seems Table 2 is not on the basis of the format of journal. Please, check all tables and figures.
We have done formatting corrections with the figures and tables according to the template. Hope they are fine now.
9- In some parts of the manuscript, paragraphing is not clear at all, for example in section 2.3 and other parts of the manuscript. Each paragraph should be started with new point and content. Please, revise it carefully.
We have revised some of the paragraphs as proposed.
10- Table 3 does not prepared on the basis of journal s format.
We have deleted some of the border lines according to the table format. Hope the format is fine now.
11- Please, check the format of reference 57.
This is a Chinese patent, we have added this information in the reference for [57].
12- All names of authors should be write for reference 56.
Done
13- n in reference 56 should big letter.
Done
14- Please, check the format of all references.
We have carefully checked the format of all references and revised them according to the journal’s guidelines.
15- Some references do not have DOI. Please, check them again that if they have DOI add it in the Reference section.
We have checked all articles that lacked DOI.
Other sections of the article is OK. The article can be accepted after considering these suggestions.
Reviewer 2 Report (Previous Reviewer 2)
Comments and Suggestions for Authors
Despite presenting only minor advances in the area, the manuscript is adequate and can be published.
Author Response
Reviewer 2:
Despite presenting only minor advances in the area, the manuscript is adequate and can be published.
We gratefully thank the reviewer for recommending the manuscript for publication.
This manuscript is a resubmission of an earlier submission. The following is a list of the peer review reports and author responses from that submission.
Round 1
Reviewer 1 Report
Comments and Suggestions for Authors
It can be accepted in present format.
Reviewer 2 Report
Comments and Suggestions for Authors
This manuscript presents a survey of plant species used locally for the treatment of mental illnesses. Collecting information on the traditional use of natural products for the treatment of diseases (ethnopharmacognosy) is a relevant field of study, but the exact objective of the present study is unclear. If the aim was to evaluate potential new treatments for these diseases, the manuscript does not present any data on efficacy, and information on pharmacognosy is quite limited.
A very worrying issue is that several species listed are recognized as toxic plants, which is not mentioned in the manuscript.
The authors mentioned in the abstract and in some other parts of the manuscript that the diseases treated would be included in the "modern concept of mental illnesses". However, the authors do not define what this "modern concept of mental illnesses" would be and do not list the specific treatment for each mental illness.
Much information requested in the questionnaire is not presented in the manuscript. There is no explanation in the manuscript about the criteria adopted to prepare the questionnaire.
As this is research involving human beings, approval of the study protocol by an ethics committee is necessary before conducting the study, even for a non-interventional study.
Reviewer 3 Report
Comments and Suggestions for Authors
The Methodology is good, some comments concerning the used terms are presented as comments in the manuscript.
Best Regards

Round 2
Reviewer 2 Report
Comments and Suggestions for Authors
My concerns with the manuscript remain the same, but I will highlight only two main topics.
One situation that worries me is that several toxic plants are listed in the manuscript. However, only three species are mentioned in the discussion as toxic, that is, most of the toxic species were disregarded. Furthermore, this paragraph of the discussion is extremely limited and superficial.
I also have some ethical considerations regarding this study. The attached questionnaire has been modified, so it is not possible to know for sure what type of questions were asked. Because the study involved interviews, approval must be obtained from an ethics committee. Previous ethics approval is required for research interviews if the information obtained from the interview is the research data. Ethics approval is not necessary only in cases where the information provided during the interview is not the subject of the research. As two of the authors are affiliated with the University of Oslo, it is mandatory to consult the Guidelines of The Norwegian National Research Ethics Committee for medical and health research. It is not a matter of opinion, as stated by the authors in the answers to the questions.